# Impacts of Psychopharmaceuticals on the Neurodevelopment of Aquatic Wildlife: A Call for Increased Knowledge Exchange across Disciplines to Highlight Implications for Human Health

**DOI:** 10.3390/ijerph18105094

**Published:** 2021-05-12

**Authors:** Stephanie J. Chan, Veronica I. Nutting, Talia A. Natterson, Barbara N. Horowitz

**Affiliations:** 1Department of Human Developmental and Regenerative Biology, Harvard University, Cambridge, MA 02138, USA; stephanie_chan@college.harvard.edu; 2Department of Computer Science, Harvard University, Cambridge, MA 02138, USA; veronicanutting@college.harvard.edu; 3Crossroads School for Arts and Sciences, 1714 21st Street, Santa Monica, CA 90404, USA; talianatterson@gmail.com; 4Department of Medicine, Harvard Medical School, Boston, MA 02115, USA; 5Department of Human Evolutionary Biology, Harvard University, Cambridge, MA 02138, USA; 6Division of Cardiology, David Geffen School of Medicine at UCLA, Los Angeles, CA 90024, USA

**Keywords:** aquatic, wildlife, toxicology, adolescent, pharmaceuticals, water

## Abstract

The global use of psychopharmaceuticals such as antidepressants has been steadily increasing. However, the environmental consequences of increased use are rarely considered by medical professionals. Worldwide monitoring efforts have shown that pharmaceuticals are amongst the multitude of anthropogenic pollutants found in our waterways, where excretion via urine and feces is thought to be the primary mode of pharmaceutical contamination. Despite the lack of clarity surrounding the effects of the unintentional exposure to these chemicals, most notably in babies and in developing fetuses, the US Environmental Protection Agency does not currently regulate any psychopharmaceuticals in drinking water. As the underlying reasons for the increased incidence of mental illness—particularly in young children and adolescents—are poorly understood, the potential effects of unintentional exposure warrant more attention. Thus, although links between environmental contamination and physiological and behavioral changes in wildlife species—most notably in fish—have been used by ecologists and wildlife biologists to drive conservation policy and management practices, we hypothesize that this knowledge may be underutilized by medical professionals. In order to test this hypothesis, we created a hierarchically-organized citation network built around a highly-cited “parent” article to explore connections between aquatic toxicology and medical fields related to neurodevelopment. As suspected, we observed that studies in medical fields such as developmental neuroscience, obstetrics and gynecology, pediatrics, and psychiatry cite very few to no papers in the aquatic sciences. Our results underscore the need for increased transdisciplinary communication and information exchange between the aquatic sciences and medical fields.

## 1. Introduction

### 1.1. Psychopharmaceuticals in Waterways across the Globe

The global use of psychopharmaceuticals to treat cognitive, behavioral, and emotional disorders has increased substantially in the last several decades [1]. In the US alone, the total number of antipsychotics purchased in the general population increased 86% from 1997 to 2007 (17.4 million to 32.4 million purchases, respectively) [2]. This precipitous increase can be partially attributed to the emergence of second-generation antipsychotics in the 1990s, as well as an increase in the number of US Food and Drug Administration (FDA)-approved conditions for which psychopharmaceuticals can be prescribed [3,4,5,6,7]. Moreover, increases in the use of psychopharmaceuticals in children, as well as off-label and non-FDA-approved uses of these medications, have further contributed to increased use [8,9,10,11,12].

Health care providers prescribe psychopharmaceuticals to prevent neurodevelopmental disorders and reduce the risk of mental illness in their patients in a wide range of medical fields (e.g., pediatrics, obstetrics and gynecology (OB-GYN), geriatrics, and psychiatry) [6,10,13]. Of the many psychopharmaceuticals consumed by humans, selective serotonin reuptake inhibitors (SSRIs), or antidepressants, are of particular interest given their widespread use and well-documented effects on neurodevelopment [14]. From 2009 to 2012, antidepressants were one of the most prescribed drugs in the US, second only to antihyperlipidemic agents (for high cholesterol) (National Center for Health Statistics (US) 2016). From 2015 to 2018, 13.2% of adults reported having used antidepressant medications in the past 30 days [15]. While sertraline and fluoxetine (i.e., Prozac) were the most commonly prescribed antidepressant medications for Americans from 1996–2015, newer SSRIs such as citalopram and escitalopram have also gained traction as treatments [16].

Despite considerable increases in the prevalence of psychopharmaceutical use, the final fate of these chemicals is often not considered beyond the point of prescription or ingestion. Monitoring efforts clearly show that pharmaceutical contaminants have joined the multitude of anthropogenic pollutants detected in the water cycle, mainly owing to the excretion of active pharmaceutical ingredients and their metabolites in urine and feces and the disposal of unwanted medications [17,18,19,20,21]. Pharmaceuticals that enter wastewater treatment plants and persist in wastewater effluents flow into surface waters, and can ultimately end up in drinking water (Figure 1) [22,23]. In addition, wastewater treatment plants (WWTPs) that receive substantial flows from pharmaceutical formulation facilities have been shown to have pharmaceutical concentrations 10–1000 times higher than typical WWTP effluents [24]. A study by Hughes et al.—which documented the extent of pharmaceutical contamination in fresh water systems on a global scale—showed that the SSRIs fluoxetine and citalopram are among the top 61 pharmaceuticals studied in waterways in over 40 countries on five continents [25]. A more recent study by aus der Beek et al.,detected a total of 631 pharmaceutical substances in 71 countries on six continents, of which the UK, Germany, Spain, and the US exhibited the highest quantities detected [26]. Naturally, this has raised concerns about the extent to which psychopharmaceutical components are ending up in waterways, where they are both unintentionally consumed by humans and are contaminating the environment [27,28,29]. 

After disposal or excretion, psychopharmaceuticals such as SSRIs make their way into wastewater treatment plants or landfills, surface and groundwater, and ultimately into human drinking water.

While pharmaceuticals have been detected in surface waters in concentrations ranging from ng/L to mg/L depending on the location and proximity to WWTP effluents [26,30] average global concentrations of commonly prescribed SSRIs in surface waters are in the range of 0.012–1.4 µg/L [31]. However, one study, which critically evaluated sampling strategies and techniques for assessing pharmaceutical contamination in waterways, determined that reported concentrations were likely not reliable or representative of actual in situ concentrations in the vast majority of studies [25], emphasizing the need for standardized strategies and methodologies for accurate assessments of contamination. 

Although the presence of pharmaceutical pollutants in US waterways has been well documented, the US Environmental Protection Agency (EPA) only regulates approximately 90 contaminants in drinking water, none of which are psychopharmaceuticals [32]. While links between environmental contamination and physiological and behavioral changes in wildlife species are used by ecologists and wildlife biologists to drive conservation policy [33], this knowledge may be underutilized by medical professionals. Thus, the aim of the present study was to test the hypothesis that an information gap exists between wildlife biology and human health care, at the intersection of ecotoxicology and medicine. 

### 1.2. Psychopharmaceuticals in Wildlife: Fish as a Model

In rodent models, commonly prescribed antidepressants such as the SSRIs fluoxetine, clomipramine, and citalopram have been shown to negatively impact neurodevelopment [34,35]. As physiological pathways tend to be well conserved, it comes as no surprise that psychopharmaceuticals also exert negative effects on neurodevelopment in wildlife [27]. Serotonin is known to be involved in a wide range of physiological and developmental processes in a diverse spectrum of aquatic organisms, which have been shown to exhibit physiological and behavioral changes in response to antidepressants and serotonin in laboratory settings [36]. For example, exposure to environmentally-relevant concentrations of SSRIs induced changes in swimming behavior in amphipod crustaceans within hours of exposure and spawning in mollusks within minutes [37]. Because serotonergic systems are evolutionarily conserved across both vertebrate and invertebrate taxa [38], the presence or absence of human drug target orthologs could be used to predict the potential effects of pharmaceuticals in a broad range of taxonomic groups [39]. 

The adverse effects of SSRIs on physiology and neurodevelopment have been particularly well studied in fish. Numerous studies have demonstrated that fish exposed to a wide range of environmentally relevant concentrations of SSRIs or other antidepressants exhibit substantial physiological and neurological changes in a sex- and dose-dependent manner [31,39,40,41,42,43,44,45]. The observed effects of antidepressants in fish include: changes in behavior (e.g., mating behavior, predator avoidance, anxiety, and aggression); substantial developmental and physiological abnormalities; changes in sexual selection, growth, and sperm count; and mortality. Some studies in fish and aquatic invertebrates show that the antidepressant fluoxetine can result in chronic effects from exposure in the low ng L^−1^ range, and thus at concentrations well below those detected in the environment [43,46]. If the observed effects of exposure to environmentally relevant concentrations of antidepressants in fish have the potential to affect neurodevelopment, reproductive success, and ultimately fitness, what then are the effects of unintentional exposure to psychopharmaceutical contaminants in humans? 

### 1.3. Antidepressants and Mammalian Development

The demonstrated adverse effects of environmentally relevant concentrations of psychopharmaceuticals on physiology and neurodevelopment in wildlife also have important implications for human development and health, which raises concerns beyond wildlife ecology and conservation. However, while the safety of incidental exposures to pharmaceuticals are often measured by the minimum therapeutic dose required to elicit a clinical response [23], perturbations at the molecular and cellular level are difficult to assess in humans in vivo. Furthermore, the effects of long-term low-level exposure to psychopharmaceuticals, the effects of exposure to multiple chemicals, and the effects of unintentional exposure to psychopharmaceuticals in babies and the fetuses of pregnant women are largely unknown [23,47].

The foundation for patterns of emotional behavior in adults are thought to be laid during embryonic development and early postnatal life. Developmental plasticity allows for windows of vulnerability, during which organisms are highly sensitive to genetic and environmental factors [48]. In mouse models, gestational exposure to SSRIs as a consequence of maternal ingestion was thought to be associated with a greater risk of adolescent depression [48], while fluoxetine exposure during early postnatal development has been linked to anxiety- and depression-like behaviors at adulthood in rodent models [49]. Thus, as serotonergic systems are known to play a central role in mammalian neurodevelopment, especially in neuroanatomical regions associated with emotional and cognitive processes, exposure to internal and external factors such as SSRIs during the development of the central nervous system puts developing fetuses at risk of brain disorders by perturbing serotonin signaling [48]. Although several studies have demonstrated changes in gene expression and nerve growth factor signaling in the placenta of pregnant women being treated with SSRIs [35,50], effects on the developing fetus are poorly understood. 

The rates of some neuropsychiatric disorders in children, adolescents, and young adults have been increasing concomitantly with increases in the use of psychopharmaceuticals. From 2003 to 2012, diagnoses of anxiety and depression in children and adolescents aged 6–17 years in the United States increased from 5.4% to 8.4% [51]. American children and adolescents seeking mental health services increased twofold, from 1.56 million youths between 1996 and 1998 to 2.28 million youths between 2010 and 2012 [52]. While a variety of environmental and genetic factors have been linked to various types of anxiety in children [53], the underlying reasons for the increased incidence of mental illness in children and adolescents remain unclear.

### 1.4. A Call for Transdisciplinary Communication

The degree to which human neurodevelopment is impacted by unintentional exposure to environmental psychopharmaceuticals is unknown (Petrie, Barden, and Kasprzyk-Hordern 2015; Wu and Janssen 2011; Hughes, Kay, and Brown 2013). In light of the expanding body of research that demonstrates clear correlations between the exposure to psychopharmaceuticals and altered neurodevelopment and behavior in wildlife, there have been calls for better governmental regulation of environmental contaminants in waterways (Wu and Janssen 2011). However, it is unclear whether information related to environmental exposures to these agents is reaching health professionals that are actively engaged in the diagnosis, treatment, and prevention of neuropsychiatric disorders.

We propose that there are knowledge and communication gaps between wildlife biologists and human health experts, which is especially evident with regards to the potential effects of environmental exposure to psychopharmaceuticals on human neurodevelopment. In order to assess whether this gap exists, we generated a hierarchically organized citation network to visualize the exchange of information between fields of study. The visualization created here illustrates a startling gap in communication and information exchange between journals of different disciplines.

## 2. Methods

### 2.1. Selection of Parent Article

The Web of Science (WoS) database was used to determine the most highly-cited journal article related to the topic of psychopharmaceuticals altering behavior in aquatic wildlife. A set of search terms were inputted into the WoS on 8 August 2020 using the following parameters: “Advanced search” for “All databases” and “All years (1864–2020)”. The search sequence and terms are shown in Table 1. While human exposures to different environmental chemicals, antimicrobials, endocrine disrupting chemicals (EDCs), and selective serotonin reuptake inhibitors (SSRIs) were included in the original search sequence, the selection criteria were honed to focus the research specifically on SSRIs. The search terms and sequence returned a total of 413 entries in WoS. The WoS “times cited” filter was then applied to screen for the top-cited journal articles. Abstracts were then manually scanned for relevance to our research focus. The top-cited article that met the following criteria was chosen as the “parent” article for the hierarchically organized citation network: SSRIs + aquatic organisms + in situ wild conditions + environmentally relevant levels of contaminants. Given our focus on “in situ wild conditions”, review articles were excluded from the selection of the parent article. Screening for these criteria led to the selection of the article “Environmental concentrations of the selective serotonin reuptake inhibitor fluoxetine impact specific behaviors involved in reproduction, feeding and predator avoidance in the fish *Pimephales promelas* (fathead minnow)” [31], hereafter referred to as the “parent” article according to the terminology of Holten [54]. 

### 2.2. Hierarchically Organized Citation Network

All articles that directly cited the parent article will be herein referred to as the “child” articles according to Holten [54]. All articles that directly cited the child articles will be likewise referred to as the “grandchild” articles. The WoS full citation records (containing article titles, authors, source title, etc.) for all child and grandchild articles were downloaded, compiled, and reorganized in a spreadsheet. Although “All Databases” was queried initially, a small number of non-WoS articles could not be accessed or retrieved. Hence, only WoS Core Collection citations were included in the hierarchically organized citation network analysis. The final dataset for the citation network included one parent article, 115 child articles, and 1318 grandchild articles. 

### 2.3. Database for Journal Categorizations 

The SCImago Journal and Country Rank Database (https://www.scimagojr.com/journalrank.php) (accessed on 11 March 2021) was used to categorize the journals of the parent, child, and grandchild articles into one or more thematic areas and subject categories.

#### Selecting Categories for Comparison

The parent article is of considerable importance to a number of fields related to human health. These fields include developmental health (i.e., obstetrics and gynecology), neurodevelopment, pediatrics, psychiatry, public health, and health policy. The following SCImago thematic areas and subject categories for journals were therefore selected for analysis: (1) Medicine: “Psychiatry and Mental Health”; (2) Medicine: “Pediatrics, Perinatology, and Child Health”; (3) Medicine: “Obstetrics and Gynecology”; (4) Neuroscience: “Developmental Neuroscience”; (5) Medicine: “Public Health, Environmental and Occupational Health”; and (6) Environmental Science: “Management, Monitoring, Policy, and Law”. The SCImago thematic area and subject category (7) Agricultural and Biological Sciences: “Aquatic Science” was also selected and closely examined given its relevance to research on the effects of pharmaceuticals in fish.

The full table of journals listed under each of the aforementioned subject categories of interest was downloaded. The following were listed on the SCImago website as of December 2020: 545 Psychiatry and Mental Health journals; 310 Pediatrics, Perinatology and Child Health journals; 185 Obstetrics and Gynecology journals; 37 Developmental Neuroscience journals; 559 Public Health, Environmental and Occupational Health journals; 349 Management, Monitoring, Policy, and Law journals; and 230 Aquatic Science journals. 

### 2.4. Assigning Journals to SCImago Subject Categories

A Python script was written to cross-compare the journals of the parent, child, and grandchild articles to the journals of the seven aforementioned SCImago subject categories (Appendix A). The code works as follows: The Python program designed here takes in the full citation records of the parent and stores a set of all distinct journal titles represented at the parent, child, and grandchild levels. Then, for each subject category of interest, the full table of journals from SCImago that corresponds to the subject category is read in and stored as a set. The intersection of the subject category set with the set of all distinct journal titles is then computed for each subject category of interest. The results of this series of set intersections are stored and can be used to indicate which journals in the citation network of the parent article fall under each of the SCImago subject categories of interest. 

To address situations in which a journal falls under multiple SCImago subject categories of interest, a prioritization scheme was developed based on the clinical relevance of each subject category. Categories were numbered from 1 to 7 as denoted in the previous section. Under this prioritization scheme, subject category (1) is prioritized over category (2), category (2) is prioritized over category (3), and so on and so forth. Under this prioritization scheme, for example, a journal that fell under both “Psychiatry and Mental Health” (1) and “Obstetrics and Gynecology” (3) would be classified as a “Psychiatry and Mental Health” journal. Ultimately, there were only two collisions in this journal classification process, in which two journals appeared under “Management, Monitoring, Policy, and Law” (6), and “Aquatic Science” (7). As per the prioritization scheme, these were both classified as “Management, Monitoring, Policy, and Law” (6) journals.

After this code was run for all seven SCImago categories of interest, remaining journals that did not fall into any of the Categories of interest were categorized as “Other”. All subject categories except “Obstetrics and Gynecology” and “Pediatrics, Perinatology, and Child Health” produced at least one intersection in our network, indicating that at least one journal under that subject category was represented in our network. Ultimately, each of the 301 distinct journals in our network was categorized as either one of the seven subject categories of interest or as “Other”. 

### 2.5. Calculating Edge Weights in Network Graph

Another Python script was written to read in the full list of citations in the network and output a graphical representation of the network. Subject categories were represented as nodes and the citations between them as directional edges. The edge weights represent the number of citations between pairs of subject categories. For example, the directional edge (i.e., arrow) from “Developmental Neuroscience” to “Aquatic Science” has an edge weight of two, indicating there are two “Developmental Neuroscience” to “Aquatic Science” citation instances in the network.

### 2.6. Creation of Citation Network Visualization

Kumu (https://kumu.io) (accessed on 11 March 2021) was used to build the citation network visualization (Figure 2). Link and node data were imported directly into the Kumu map. The size of the nodes and thickness of the links were set to be proportional to the number of articles in a particular subject category and the number of articles cited between pairs of subject categories, respectively. Arrows indicate directionality. The direction of the links was determined by the direction of the citation: from the citer to the cited.

## 3. Results

Figure 2 tracks and visualizes the flow of citations across disciplines, and effectively illustrates the extent to which information is being transferred into human health-related literature. In this study, we analyzed the citation network of a single, strategically-chosen study (see Methods), the parent, which details psychopharmaceutical-induced changes in wildlife behavior (Figure 2A). We identified a total of 1433 citations across the child and grandchild-levels of our citation network. The citations were categorized and visualized as directional edges between nodes. 

Since the parent study was published in an “Aquatic Science” journal, we closely examined links coming into and out of the “Aquatic Science” node. The knowledge and communication gaps between wildlife biologists and human health experts is most robustly illustrated by the paucity of citations between certain subject categories. Two health-related subject categories of interest are completely unrepresented in our network: “Obstetrics and Gynecology” and “Pediatrics, Perinatology, and Child Health,” indicating that no OB-GYN or pediatric articles cited the parent article or its children. Thus, research from our parent article is not flowing into the OB-GYN or pediatric fields. Our results also show that communication between “Aquatic Science” and other developmental health-oriented fields was also sparse. Of the 1433 total citations in the network, only two “Developmental Neuroscience” articles, two “Psychiatry and Mental Health” articles, 16 “Public Health, Environmental and Occupational Health” articles, and 15 “Management, Monitoring, Policy, and Law” articles cited “Aquatic Science” articles. Thus, information from “Aquatic Science” articles in our network was predominantly cited by articles in non-developmental scientific fields (i.e., “Other”). 

## 4. Discussion

Our findings confirm that there is indeed an absence of communication between wildlife biologists and human health experts concerning the potential health effects of exposure to psychopharmaceuticals in our waterways. Most strikingly, our results demonstrate that there is little to no interdisciplinary dialogue between aquatic science and the fields of pediatrics and OB-GYN, underscoring the importance of bridging this gap. 

Studies have demonstrated fundamental similarities across vertebrate taxa in neurodevelopment and vulnerability to toxic exposures [55]. The well-documented effects of antidepressants in fish (i.e., changes in reproductive success, behavior and hormones, mass and body condition, and activity levels) are analogous to the side effects of antidepressants in humans (i.e., sexual dysfunction, anxiety and suicidal thoughts, weight loss, and feeling restless, respectively) [27]. Furthermore, the brain morphology of zebrafish has been described as “stinkingly similar” to that of mammalian (rodent) models, from the macro to the cellular level [55]. Thus, the effects of psychopharmaceuticals on neurodevelopment in wild fish are directly relevant to human neurodevelopment, as well as vulnerability to psychopathology in later life. 

The existing literature suggests that pharmaceuticals are generally present in our waterways at subtherapeutic levels [25]; however, as psychopharmaceuticals are designed to have biological effects at low doses, and environmental contamination often consists of multiple active pharmaceutical ingredients [22,27], the chronic, long-term exposure to these chemicals warrants more investigation in both aquatic biota and humans alike. In a critical review of the literature demonstrating the effects of chronic exposure to pharmaceuticals in aquatic biota using both lethal and sublethal indices, Hughes et al., noted that “antidepressants appear to pose particular risk to all taxa except bacteria”, further underscoring the importance of this issue [25]. 

As human and animal environments increasingly overlap, ecological contaminants are disturbing a broad range of species [56,57]. We are all familiar with the canary in the coal mine, and thus the idea that animal species can serve as potential indicators of environmental hazards and impending threats to public health is not new [57]. While the impact of pharmaceutical contamination on animal behavior is of particular concern to ecologists and conservation biologists [28,29,31,42,45,58], we postulate that equal concern should be placed on the potential effects of these substances on human neurodevelopment, and their relationship to biobehavioral disturbances and psychiatric conditions. In addition, we believe that the adverse effects of environmental pollutants on aquatic organisms—most notably in fish—are extremely relevant to a wide range of human health professionals (e.g., physicians, scientific investigators, child psychiatrists, policymakers, and public health officials concerned with gestational health and neurodevelopment). However, the extent to which human health professionals are aware of the adverse effects of environmental pollutants in wildlife is not clear. As medical fields become increasingly specialized, information becomes more compartmentalized and siloed, creating even more barriers to the exchange of transdisciplinary knowledge, among other problems. In the case of environmental contamination and its impact on both humans and wildlife, this lack of knowledge exchange could be creating a blind spot that obscures essential aspects of the issue. 

While our initial citation network highlights the absence of transdisciplinary knowledge exchange, we recognize that the citation data stem from one relatively recent study. Future studies should analyze the citation networks of more articles for a more complete picture. In addition, our current understanding of the effects of exposure to psychopharmaceutical residues in waterways on neurodevelopmental is also limited by the minimal research regarding this potential hazard. Lastly, as the main fields covered in the present study are constantly evolving, more articles have cited the parent article since the citation map was created on 8 August 2020. Further studies could document the evolution of citations to assess whether the knowledge gaps are closing. 

At a minimum, the levels of psychopharmaceuticals in waterways should be subject to regulation, and the results made readily available to the public in an accessible and transparent manner. Beyond the cadre of studies that focus on the effects of psychopharmaceuticals in wildlife, further research should be conducted to assess real-life incidental exposures in humans of all ages. Above all, there should be a greater awareness of the value of natural animal models as a resource to better understand the influence of environmental contaminants on human neurodevelopment. Effecting the necessary cultural shift in how disciplines engage with each other will require an integrated and multi-pronged approach. Among the interventions that can accelerate this change are: (1) support for interdisciplinary conferences and research collaborations focused on the effects of the environment on development across species, (2) enhanced environmental health instruction at all levels of medical education, and (3) encouraging the editorial leadership of widely-read clinical medical journals to increase the publication of ecologically and environmentally focused research with relevance to human health and development.

As our case study shows, the traditional method for disseminating information does not presently provide a sufficient exchange of knowledge across disciplines. Increasing knowledge exchange has the potential to alert both physicians and patients to the rapidly growing body of evidence that links pharmaceutical and other environmental contaminants to every area of human health. 

## Figures and Tables

**Figure 1 ijerph-18-05094-f001:**
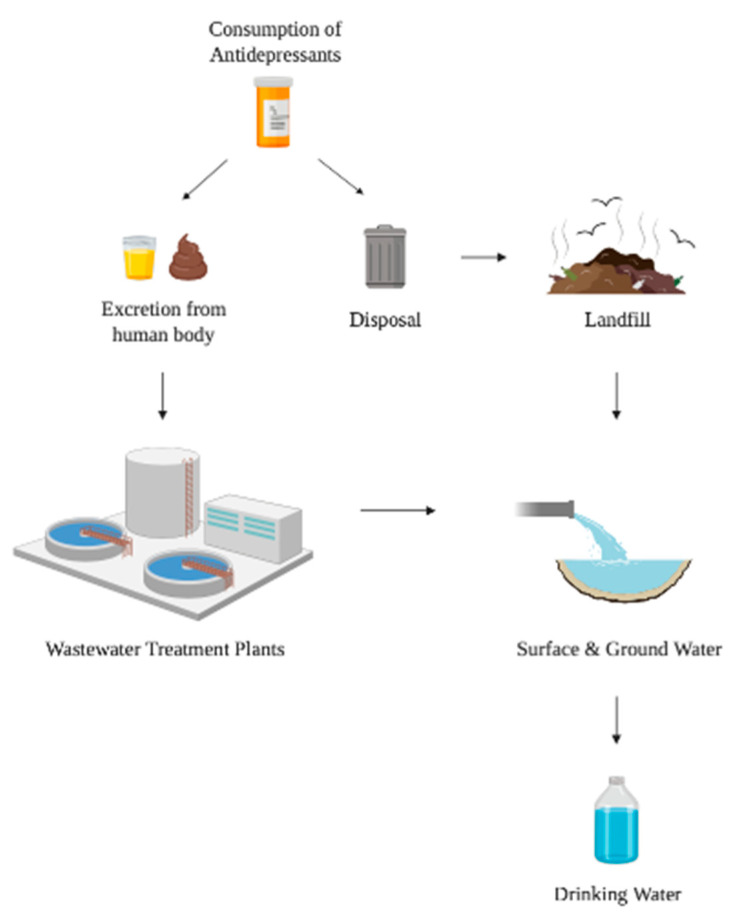
The path of psychopharmaceuticals into drinking water.

**Figure 2 ijerph-18-05094-f002:**
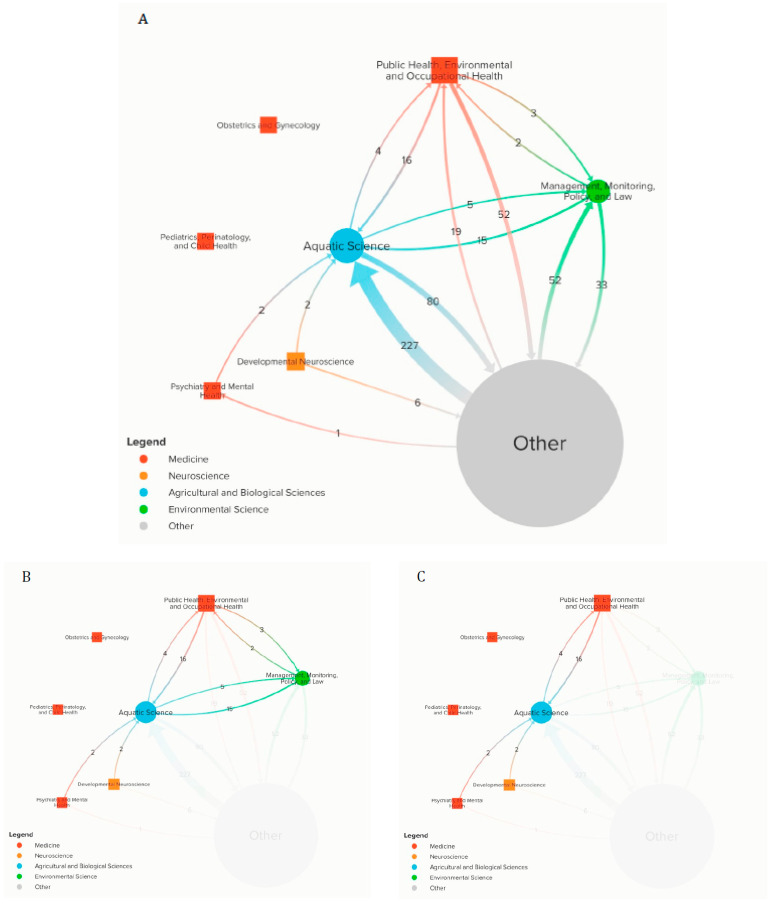
Citation Network for Weinberger and Klaper Article [31]. (**A**) Citation network of articles belonging to non-developmental scientific fields and the seven selected SCImago Categories of interest: (1) Psychiatry and Mental Health; (2) Pediatrics, Perinatology and Child Health; (3) Obstetrics and Gynecology; (4) Developmental Neuroscience; (5) Public Health, Environmental and Occupational Health; (6) Management, Monitoring, Policy, and Law; and (7) Aquatic Science. “Other” refers to articles that did not fall into any of the seven SCImago categories of interest, and citations from these articles appear as arrows coming out of this node. (**B**) Citation network of articles belonging to only the aforementioned seven SCImago categories of interest. (**C**) Citation network of articles belonging to Aquatic Science and the clinically relevant categories of Psychiatry and Mental Health; Pediatrics, Perinatology, and Child Health; Obstetrics and Gynecology; and Developmental Neuroscience. The size of the nodes and thickness of the links were set to be proportional to the number of articles in a particular Category and the number of articles citing between pairs of Categories, respectively. Arrows indicate directionality. The direction of the links was determined by the direction of the citation: from the citer to the cited.

**Table 1 ijerph-18-05094-t001:** Search sequence entered into the Web of Science “Advanced search” for “All databases” and “All years (1864–2020)”. The terms were based on parameters aimed at identifying papers that featured alterations in wildlife behavior in response to pharmaceuticals.

Set	Exact Search Terms	Results
1	TS = (“selective serotonin reuptake inhibitor” OR “antimicrobial” OR “endocrine disrupting chemical”)	461,483
2	TS = behavior	7,123,816
3	TS = (mammal OR reptile OR bird OR amphibian OR fish)	23,978,264
4	TS = (transgenic OR laboratory OR rodent OR zebrafish OR “model organism”)	10,681,413
5	1 AND 2 AND 3	11,538
6	5 NOT 4	7593
7	TS = (human)	29,475,987
8	TS = (“antimicrobial peptide”)	39,031
9	TS = (pig OR porcine OR cow OR bovine OR chicken OR poultry)	3,733,404
10	6 NOT 7 NOT 8 NOT 9	413

TS = topic.

## Data Availability

Not applicable.

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
