# Peer review of "Impacts of Psychopharmaceuticals on the Neurodevelopment of Aquatic Wildlife: A Call for Increased Knowledge Exchange across Disciplines to Highlight Implications for Human Health"

_ijerph, 2021, doi:10.3390/ijerph18105094_

Round 1
Reviewer 1 Report
This is a well-written and compelling piece of research showing the lack of citations between fisheries research and public health research in regard to psychopharmaceutical pollution in waterways. The methods are good and presented well. The topic is perfectly matched to the scope of the journal. I have one minor suggestion: The title is a bit long and vague. I suggest referencing psychopharmaceuticals in the title and making it more direct. Something like: Medical fields can learn from fish about potential impacts to human health from psychopharmaceutical pollution in our waterways.
Author Response
Thank you very much for your comments and suggestion. We have reworded the title in the revised version of the manuscript.
Reviewer 2 Report
This article deals with a very interesting and current topic, which deserves great attention globally. It emphasizes the necessary transdisciplinary communication and information exchange between the aquatic sciences and medical fields since, as herein demonstrated, it is still almost totally missing though its urgent and utmost importance for the environment and human health. In particular, concern on the increased use of psychopharmaceuticals globally is even worsen by the current Covid-19 pandemic, which maybe authors should briefly mention in their text.
In my opinion, the quality of the paper is very good, both in terms of originality, structure, English grammar and article search and correlation. Only, I am not sure if it is proper to considered this text as an Article, as maybe it would fit better as a critical Review or maybe as a Research note. However, few comments are indicated below for a minor revision of this manuscript.
In the title, please check punctuation. I think that the second sentence must be split in two phrases.
Even if of general knowledge, all abbreviations used in the text must be fully described at their first mention. Please provide the meaning of EPA in the abstract, FDA in Introduction, and so on throughout the entire text of the manuscript. After first definition, authors may then use abbreviation.
Almost at the end of the first paragraph of Introduction, pag. 3, what the authors mean with “the presence of pharmaceutical pollutants in our waterways...”? Does “our” refer to a specific geographical area? Please clearly specify, here and elsewhere in the text.
At pag. 4, World Health Organization after its first mention should be abbreviated, to then use lately its abbreviation. Also, at the end of this paragraph, the last sentence must be reworded as “exposure” and “effects” appear too many times, and should be replaced by other similar terms.
Some entries in Table 1 must be checked for inaccuracies or misspellings.
Author Response
Thank you for your comments and careful assessment of our paper. Please find our responses to our comments in italics below:
This article deals with a very interesting and current topic, which deserves great attention globally. It emphasizes the necessary transdisciplinary communication and information exchange between the aquatic sciences and medical fields since, as herein demonstrated, it is still almost totally missing though its urgent and utmost importance for the environment and human health. In particular, concern on the increased use of psychopharmaceuticals globally is even worsen by the current Covid-19 pandemic, which maybe authors should briefly mention in their text.
Thank you for your comments and for this astute suggestion, but as we do not have data on the increase in pharmaceutical use as a result of the Covid-19 pandemic, we have decided not to include this information in the manuscript. In addition, we feel that it is beyond the scope of the present study.
In my opinion, the quality of the paper is very good, both in terms of originality, structure, English grammar and article search and correlation. Only, I am not sure if it is proper to considered this text as an Article, as maybe it would fit better as a critical Review or maybe as a Research note. However, few comments are indicated below for a minor revision of this manuscript.
Thank you for suggesting alternative article types. We are open to which ever article type that the editors feel is best.
In the title, please check punctuation. I think that the second sentence must be split in two phrases.
We have reworked the title in the revised version of the manuscript.
Reviewer 3 Report
The paper is well written and I really enjoyed reading it. There are several concerns, but I think the main message is a useful one.
- When I searched:
TS = (transgenic OR laboratory OR rodent* OR zebrafish* OR “model organism*”) I got 1,358,841 not 10,681,413 – that required to search all databases – so the numbers in Table 1 should reflect the search that was used. Once the paper was selected the search strings for other fields should have been restricted to “since” original paper.
- It is unfortunate that the analysis was not replicated. The discussion suggests more analyses are needed but once the scripts are written it should be relatively easy to replicate the study. It is not that surprising that medical professionals would not read a fish toxicity article but perhaps review articles or environmental chemistry articles would be more likely to be cited. Searching “fluoxetine toxicity fish” finds 14 more highly cited articles, including a review article with 300 citations.
- Grandchildren of a relatively recent article difficult is difficult given that the median publishing year of “children” is 2018.
- An analysis could also look at these data in several additional ways to explore further
- how many articles from mammalian were cited in the original parent paper
- how often a paper on SSRIs and human were cited in the parent
- comparative study of rat toxicity and how often cited in aquatic
- it is not clear from the figure how many citations were in the study for potential matching – I think it is the 1433 citations. Are those the potential matches in Figure 2. How many of the 500+ citations in Figure 2 are unique?
Style and typos
- Reference style unclear why the references are numbered but cited as author, year
- Reference style inconsistent – sometimes multiple authors are “et al.” and sometimes written out (eg. P5 lines 14, 15)
- P 5 line 14 change “pharmaceutical” to pharmaceuticals
Author Response
Thank you for your insightful comments. Please find our responses to your comments below in italics.
- When I searched:
TS = (transgenic OR laboratory OR rodent* OR zebrafish* OR “model organism*”) I got 1,358,841 not 10,681,413 – that required to search all databases – so the numbers in Table 1 should reflect the search that was used. Once the paper was selected the search strings for other fields should have been restricted to “since” original paper.
Thank you for bring this lack of clarity to our attention. Please note that we get a similar number (1,375,732) when searching only the WoS Core Collection. However, searching “All Databases” returns 11,093,942 results, as more papers have been published since our original search in August of 2020. Please see sets #4 and #11 and the associated results in attached image.
2. It is unfortunate that the analysis was not replicated. The discussion suggests more analyses are needed but once the scripts are written it should be relatively easy to replicate the study. It is not that surprising that medical professionals would not read a fish toxicity article but perhaps review articles or environmental chemistry articles would be more likely to be cited. Searching “fluoxetine toxicity fish” finds 14 more highly cited articles, including a review article with 300 citations .We have to defend our choices to the reviewer. What were the criteria for selecting the parent paper?
Thank you for raising these concerns. While a more specific search would have likely returned more highly-cited articles, we developed our table of more broad search sequence terms in order to catch as many articles as possible. This is why we used the more broad category of “selective serotonin reuptake inhibitor” instead of searching for something more specific, such as “fluoxetine toxicity fish”. In addition, searching “fluoxetine toxicity fish” would limit the potential for other SSRIs to turn up in search results. In addition, we also wanted to be sure to include studies that investigated wild species (i.e., not laboratory stock) and experimentally-relevant levels of SSRIs. Lastly, because of our focus on “in situ wild conditions” we intentionally excluded review articles as the parent article. We have added a sentence to the Methods to clarify this: Given our focus on “in situ wild conditions”, review articles were excluded from the selection of the parent article.
3. Grandchildren of a relatively recent article difficult is difficult given that the median publishing year of “children” is 2018.
Thank you for bringing this to our attention. Yes, we understand that this is a potential limitation of the current study. However, the chosen paper was the top-cited study that me our selection criteria, which did not include publication year. We have added “relatively recent” to the limitations paragraph in the second to last paragraph of the Discussion to reflect our acknowledgement that the parent article was published relatively recently.
- An analysis could also look at these data in several additional ways to explore further.
Yes, we are in agreement and we hope to do this in the future.
5. how many articles from mammalian were cited in the original parent paper.
Thank you for this question. The parent has a total of 54 references. Therefore, there are 54 grandparent-parent citations. The categories of these grandparent-parent citations were not analyzed in this paper. The category of the parent article is Aquatic Science. Here is the breakdown of all 115 parent-to-child citations: 81 from Aquatic Science to Other, 21 from Aquatic Science to Other, 8 from Aquatic Science to Policy, 4 from Aquatic Science to Public Health, and 1 from Aquatic Science to Psychiatry. Please see the table in the attached PDF, which we created to address your questions regarding the citations in the parent article. Please note that the legend for yellow and green highlighting is at the end of the document/table.
6. how often a paper on SSRIs and human were cited in the parent.
Please see the attached table that was referenced in the previous comment.
7. comparative study of rat toxicity and how often cited in aquatic.
Thank you for this suggestion for further research. However, as the study is focused on the potential effects of pharmaceutical contamination in developing humans, we feel that this is beyond the scope of the present paper. This would be a good avenue to pursue in the future.
8. it is not clear from the figure how many citations were in the study for potential matching – I think it is the 1433 citations. Are those the potential matches in Figure 2. How many of the 500+ citations in Figure 2 are unique?
Yes, we analyzed and matched 1433 distinct parent-to-child and child-to-grandchild citations. In our visualizations, we have displayed citations between categories. Each arrow represents the number of unique citations between categories. Self-citations, that is to say citations between journals that fall under the same category, do not appear as arrows. The number of self-citations in the network is 1433 minus the sum of all the arrow values.
Style and typos
1. Reference style unclear why the references are numbered but cited as author, year.
Thank you for this comment. As the journal does not request a specific citation style upon submission, we used the style for Environmental Health Perspectives in our reference manager. The reference format that you see is the default format for this style
2. Reference style inconsistent – sometimes multiple authors are “et al.” and sometimes written out (eg. P5 lines 14, 15).
For this reference style, up to the first three authors are included in the in-text citations, after which “et al.” is used. We acknowledge that this is a somewhat unconventional style.
3. P5 line 14 change “pharmaceutical” to pharmaceuticals
Thank you for bringing this to our attention. This has been rectified.

Reviewer 4 Report
Dear authors,
very interesting paper. Only a few comments. Your message is very clear and important to call for more transdisciplinary research and communication. Can you give some options to achieve such a "acting beyond the own boundaries"? The only point to be improved might be to give more information on the category "other".
Best regard,
the reviewer
Author Response
Please find our responses to your comments in italics below:
Dear authors,
very interesting paper. Only a few comments. Your message is very clear and important to call for more transdisciplinary research and communication. Can you give some options to achieve such a "acting beyond the own boundaries"? The only point to be improved might be to give more information on the category "other".
Thank you very much for your comments and insightful suggestions. We have revised the last two paragraphs of the paper to provide potential avenues for increasing knowledge exchange between disciplines.
“At a minimum, the levels of psychopharmaceuticals in waterways should be subject to regulation, and the results made readily available to the public in an accessible and transparent manner. Beyond the cadre of studies that focus on the effects of psychopharmaceuticals in wildlife, further research should be conducted to assess real-life incidental exposures in humans of all ages. Above all, there should be a greater awareness of the value of natural animal models as a resource to better understand the influence of environmental contaminants on human neurodevelopment. Effecting the necessary cultural shift in how disciplines engage with each other will require an integrated and multi-pronged approach. Among the interventions that can accelerate this change are: 1) support for interdisciplinary conferences and research collaborations focused on the effects of the environment on development across species, 2) enhanced environmental health instruction at all levels of medical education, and 3) encouraging the editorial leadership of widely-read clinical medical journals to increase the publication of ecologically and environmentally focused research with relevance to human health and development.
As our case study shows, the traditional method for disseminating information does not presently provide a sufficient exchange of knowledge across disciplines. Increasing knowledge exchange has the potential to alert both physicians and patients to the rapidly growing body of evidence that links pharmaceutical and other environmental contaminants to every area of human health.”
In addition, we have also clarified the category “Other”, both within the text as well as within the caption of Figure 2.